**Data Availability Statement:** Data was obtained as a public dataset in the BMMR2 Challenge. The full

# Machine learning with textural analysis of longitudinal multiparametric MRI and molecular subtypes accurately predicts pathologic complete response in patients with invasive breast cancer

**Aaquib Syed[1], Richard Adam[1], Thomas Ren[1], Jinyu Lu[2], Takouhie Maldjian[1], Tim Q. Duong🔴[1] ***

**1** Department of Radiology, Montefiore Health System and Albert Einstein College of Medicine, Bronx, New York, United States of America, **2** Oncology Division, Department of Medicine, Montefiore Health System and Albert Einstein College of Medicine, Bronx, New York, United States of America

\* Tim.duong@einsteinmed.org

## Abstract

### Purpose

To predict pathological complete response (pCR) after neoadjuvant chemotherapy using extreme gradient boosting (XGBoost) with MRI and non-imaging data at multiple treatment timepoints.

### Material and methods

This retrospective study included breast cancer patients (n = 117) who underwent neoadjuvant chemotherapy. Data types used included tumor ADC values, diffusion-weighted and dynamic-contrast-enhanced MRI at three treatment timepoints, and patient demographics and tumor data. GLCM textural analysis was performed on MRI data. An extreme gradient boosting machine learning algorithm was used to predict pCR. Prediction performance was evaluated using the area under the curve (AUC) of the receiver operating curve along with precision and recall.

### Results

Prediction using texture features of DWI and DCE images at multiple treatment time points (AUC = 0.871; 95% CI: (0.768, 0.974; p<0.001) and (AUC = 0.903 95% CI: 0.854, 0.952; p<0.001) respectively), outperformed that using mean tumor ADC (AUC = 0.850 (95% CI: 0.764, 0.936; p<0.001)). The AUC using all MRI data was 0.933 (95% CI: 0.836, 1.03; p<0.001). The AUC using non-MRI data was 0.919 (95% CI: 0.848, 0.99; p<0.001). The highest AUC of 0.951 (95% CI: 0.909, 0.993; p<0.001) was achieved with all MRI and all non-MRI data at all time points as inputs.

dataset is available at (https://doi.org/10.7937/TCIA.KK02-6D95).

**Funding:** The author(s) received no specific funding for this work.

**Competing interests:** The authors have declared that no competing interests exist.

## Conclusion

Using XGBoost on extracted GLCM features and non-imaging data accurately predicts pCR. This early prediction of response can minimize exposure to toxic chemotherapy, allowing regimen modification mid-treatment and ultimately achieving better outcomes.

## Introduction

Neoadjuvant chemotherapy (NAC) [1] in the setting of locally advanced breast cancer can reduce tumor size, making breast conservation surgery feasible and obviating the need for mastectomy. Pathological complete response (pCR) is a desirable endpoint of NAC entailing no residual invasive tumor is present at surgery post NAC [2, 3]. Patients with pCR are more likely to be candidates for breast-conserving surgery and to have longer progression-free and overall survival [2, 3]. Therefore, pCR can be used as a surrogate for favorable outcome. The ability to predict pCR prior to treatment would help in determining which patients will benefit from NAC and which will not. Furthermore, predicting pCR can allow for changes in treatment regimens, maximizing the chances of pCR. Thus, the accurate prediction of pCR could help in identifying patients who are likely to respond to specific NAC drugs while enabling oncologists to alter treatments mid-course if needed in order to maximize successful outcomes while minimizing the adverse effects of unnecessary chemotherapy.

Currently, the efficacy of a chemotherapy regimen is tested invasively through core needle biopsy. Clinical biomarkers, such as Ki67, give a limited assessment of the entire tumor as they are obtained by core needle biopsy and therefore may not be representative of the entire tumor. Noninvasive imaging can overcome this problem of tumor heterogeneity, as the entire tumor is depicted on images [4]. In fact, the capacity of pre and post NAC MRI to depict response to treatment and predict pCR has already been demonstrated [5–7].

Machine learning (ML) has been used to predict eventual pCR. Radiomic features (such as volume, sphericity, DCE MRI signal of wash in and wash out) [8–12] and deep learning analysis of whole MR images [13], DCE dynamics [14] that include demographics and molecular receptor subtypes [15, 16] have been used to predict pCR. ML analysis has also been applied to include diffusion MRI [17–20]. However, the role of diffusion MRI to predict pCR is understudied and the results remain controversial [21].

The goal of this study was to apply the extreme gradient boosting (XGBoost) algorithm to predict pCR using multiparametric MRI data along with non-imaging data at multiple treatment timepoints as inputs. Extreme gradient boosting (XGBoost) was chosen due to the relatively small sample size of the dataset, as XGBoost has been shown to be effective even with limited data. Furthermore, XGBoost has been proven to be effective at classifying with tabular data. This approach has the potential to non-invasively identify patients who are likely to respond to neoadjuvant chemotherapy at diagnosis or early treatment. This approach may prove useful for treatment planning, treatment execution, and mid-treatment adjustment to achieve better outcomes.

## Materials and methods

### Data sources

In this retrospective study, data from the Breast Multiparametric MRI for prediction of NAC Response-2 (BMMR2) competition training dataset, which was curated from the ACRIN-6698

sub trial of the I-SPY 2 TRIAL (NCT Number: NCT01564368), was used to create machine learning models to predict pCR. Patients from the ACRIN-6698 multicenter trial were previously reported in a paper by Partridge et. Al titled "Diffusion-weighted MRI Findings Predict Pathologic Response in Neoadjuvant Treatment of Breast Cancer: The ACRIN 6698 Multicenter Trial", published in Radiology. Data collected in that trial was published as an open dataset. Our paper utilizes that data (n = 117 patients, of which 36 had pCR). While the previous paper attempted to predict pCR solely using DWI MRI with logistic regression as the model technique, we use extreme gradient boosting with multiple types of MRI data as well as patient demographic data to predict pCR. As the data containing the training and testing sets came from a public dataset, no IRB was required.

All 117 female patients in the dataset were diagnosed with invasive breast cancer and underwent 12 weeks of paclitaxel followed by 4 weeks of anthracycline treatment. Each patient sample contained collections of MRI images taken at 3 distinct timepoints, namely, tp0: pre-treatment, tp1: 3 weeks post paclitaxel (± experimental agent), and tp2: 12 weeks post paclitaxel (± experimental agent).

Non-imaging data included patient age, race, lesion type (one of multiple masses, single mass or non-mass enhancement), hormone receptor status (hormone receptor (HR) positive/negative, human epidermal growth factor receptor 2 (HER2) positive/negative), and Scarff-Bloom-Richardson (SBR) grade. Of the 117 patients, 12 patients had missing data (11 had an unknown race, and 1 had an unknown SBR grade). These values were filled using backward fill, which populates the missing data with the data from the next patient. There were no other missing data (**Fig 1**).

MRI data, including DWI and DCE MRI, were performed bilaterally in the axial orientation using a 1.5- or 3.0-T field strength magnet and a dedicated breast radiofrequency coil as described in [22]. Acquisition parameters are provided in the data source in our data availability statement. DCE images were aligned with their corresponding rectangular tumor masks with position metadata, which highlighted the tumor and the surrounding area. Textural features were extracted using the 3rd dynamic DCE data for all 3 time points in the rectangular region of interest mask. DWI images were aligned with manually defined binary segmentation data provided in the dataset. Features were calculated in the smallest rectangle that encompassed the segmented tumor using $b = 800$ s/mm$^2$ DWI imaging data. ADC tumor segmentations were carefully aligned with their corresponding ADC map with position metadata. Tumor ADC values, and changes in these values, were extracted for ML analysis. All images were interpolated to 0.7825 mm x 0.7825 mm voxel spacing with a 2.5 mm slice thickness, which was the median spacing and thickness for all patients. When applying machine learning techniques on multicenter data, data harmonization [23] may be necessary. However, we did not perform additional data harmonization as all MRI acquisitions were taken with ISPY2 acquisition requirements as described in [22]. Furthermore, we ensured that data from different field strengths and different molecular subtypes, etc., were not over-represented in either training and testing data sets by using 5-fold cross validation so that all data has the chance to be in both the test and training sets.

## Ground truth

pCR determination was done using histopathologic analysis at study sites by institutional pathologists (blinded to functional tumor volume (FTV) and ADC MRI measures) according to the I-SPY 2 trial protocol using the residual cancer burden system. Following U.S. FDA rationale and guidelines, pCR was the reference standard for determining response to neoadjuvant chemotherapy in our study, defined and reported as no residual invasive disease in either

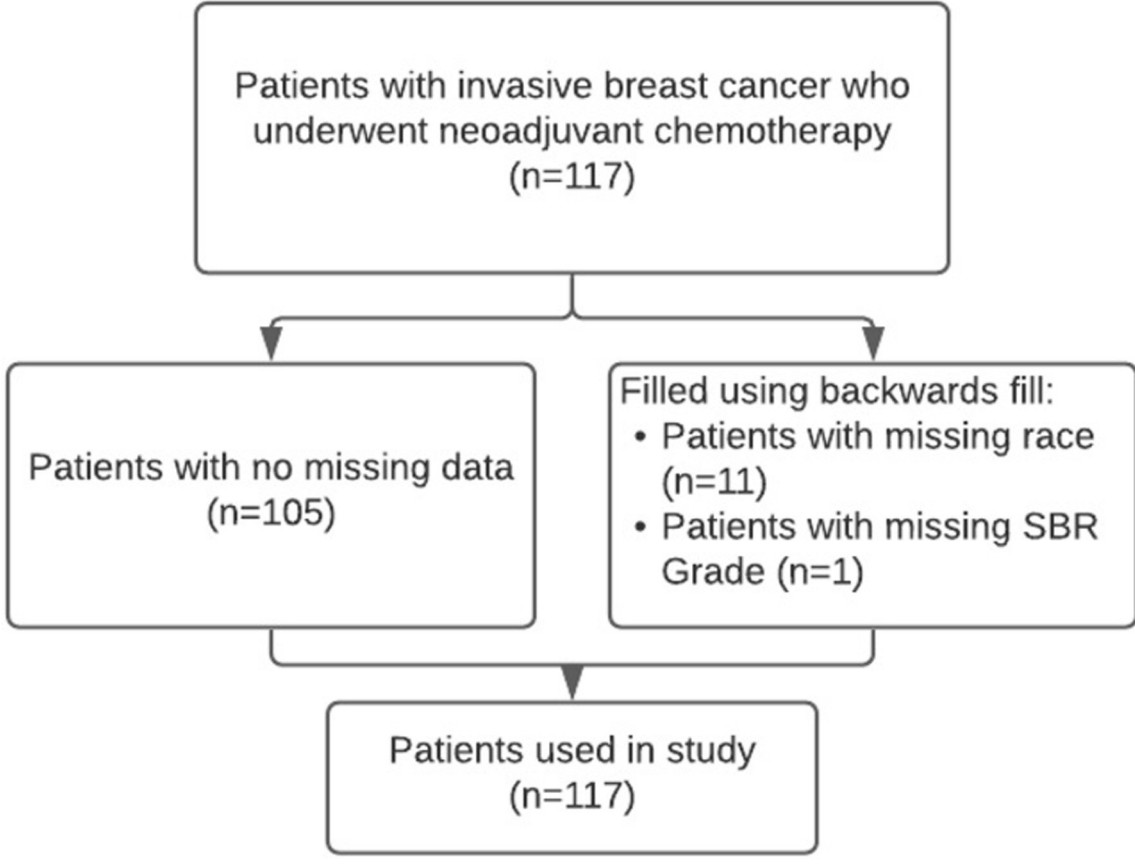

**Fig 1. Study population characteristics.**

breast or axillary lymph nodes after neoadjuvant therapy. Patients were categorized as having pCR or non-pCR based on postsurgical histopathologic examination findings.

## Features used

Texture analysis was performed using a small bounding box enclosing the tumor as determined by functional tumor volume provided by ISPY-2 data. Calculations were performed using the Scikit-Image library in Python. Images were cast to 8-bits, and the number of bins was set to the maximum value of 256 to maximize the number of grey levels counted. GLCM features (Energy, Homogeneity, Contrast, Dissimilarity, ASM, and Correlation) were calculated at distances of 1, 3, and 5 pixels at angles of 0, $\frac{\pi}{2}$, $\frac{\pi}{4}$, and $\frac{7\pi}{4}$ radians, representing all cardinal and ordinal directions. These features were calculated for all 3 treatment time points for both DCE and DWI MRI images, except for the 5-pixel direction for DWI images, as sometimes the segmentation was too small. **Table 1** summarizes the significance of each feature and the formula used to calculate it. This resulted in a total of 360 GLCM features. Combined with 6 non-imaging patient data features, and 6 features extracted from ADC parametric map, the total number of features used was 372. Feature selection to reduce the number of features was unnecessary, as XGBoost inherently selects for the most important features when splitting leaves of decision trees, automatically ignoring irrelevant features.

**Table 1. GLCM textural features used.**

| GLCM Feature Name | Formula | Purpose |
|---|---|---|
| Contrast | $\sum_{i=0}^{N-1}\sum_{j=0}^{N-1}(i-j)^2 P(i,j)$ | A measure of the intensity difference between a pixel and its neighbor, 0 for a constant image. |
| Energy | $\sum_{i=0}^{N-1}\sum_{j=0}^{N-1}p\,(i,j)^2$ | A measure of the sum of squared elements in the GLCM, 1 for a constant image. |
| Correlation | $\sum_{i=0}^{N-1}\sum_{j=0}^{N-1}\frac{(i-\mu_i)(j-\mu_j)P(i,j)}{\sigma_i\sigma_j}$ | A measure of how correlated a pixel is to its neighbors over the whole image. |
| Homogeneity | $\sum_{i=0}^{N-1}\sum_{j=0}^{N-1}\frac{p(i,j)}{1+|i-j|}$ | A measure of the closeness of the distribution of elements in the GLCM compared to the diagonal. |
| Dissimilarity | $\sum_{x=1}^{K}\sum_{y=1}^{K}|x-y|p_{xy}$ | A measure of distance between pairs of objects (pixels) in the region of interest. |
| Angular Second Moment (ASM) | $\sum_{i=0}^{N}\sum_{j=0}^{N}p(i,j)^2$ | A measure of the uniformity of distribution of grey level in the image |

## Extreme gradient boosting models

A total of 13 XGBoost models were created using different combinations of data and time points. These XGBoost models were created in Python using the Scikit-Learn API. Minority oversampling was used to balance the frequency of each class in the data set by randomly over-sampling the minority (pCR) class. The oversampled balanced dataset consisted of 162 patients with 50% pCR and 50% non-PCR outcomes.

Bayesian optimization along with a sequential domain reduction transformer was used to find optimal values of these hyperparameters. The mean AUC after 5-fold cross-verification was selected as the variable to maximize. 15 rounds of random exploration and 80 rounds of optimization were used. If the optimal value was an extreme of the bound, the bounds were adjusted, and optimization was run again. **Fig 2** shows this process for a sample hyperparameter.

## Statistical analysis

Distributions of patient characteristics, such as the distribution of lesion types and race, were compared using the $\chi 2$ test for homogeneity. Patient age and maximum tumor diameter

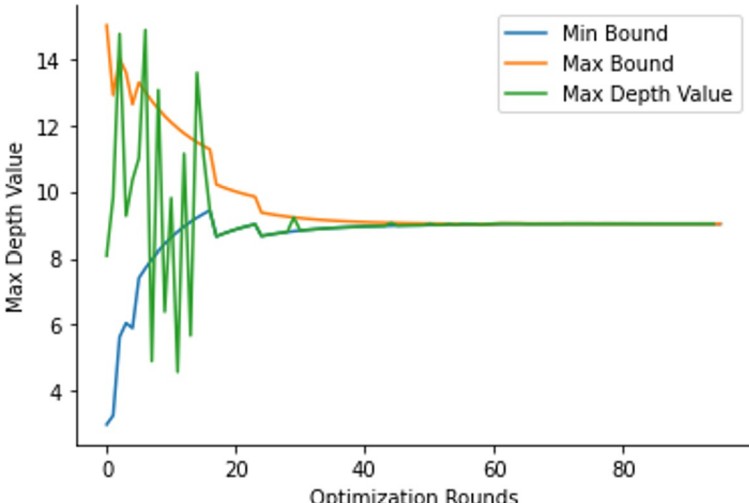

**Fig 2. Sequential domain reduction transformer on the "max depth" hyperparameter.**

distributions between classes were compared using a t-test. Investigation of the receptor status characteristics were done using 2 sample z-test of proportions. F-scores were used to calculate the importance of each individual feature in predicting pCR.

K-fold cross-validation is considered the gold standard for determining model performance after training. 5 folds of verification were used to designate 20% of the dataset as testing and 80% as training for each fold. All performance metrics were calculated as the mean value of each testing metric of the 5 folds after 1000 rounds of training. Standard error of the mean was calculated as the variability of this mean AUC.

Models were analyzed primarily through the area under the receiver operator curve (AUC). AUC has been shown to be the optimal method for comparing AI models. Precision and recall were secondary metrics for model performance. P-values < 0.05 were considered significant and were calculated as the probability that the null hypothesis is true.

## Results

Patient characteristics and the sample sizes (n = 117) are described in **Table 2**. There were 36 patients with pCR and 81 without pCR. There was no significant difference in the mean age for patients with pCR (49.08 +/- 10.31 years) and patients without pCR (49.0 +/- 11.73 years) (p = 0.971). There was also no significant difference in the longest diameter of the tumor between classes (p = 0.252). The distributions of patient races (p = 0.205), lesion types (p = 0.409), and SBR grades (p = 0.488) showed no significant difference. There was a significant difference in distributions of receptor statuses (p<0.001), with the patient without pCR

**Table 2. Patient demographics and sample sizes.**

|  | Patients with pCR (n = 36) | Patients without pCR (n = 81) | P-Value |
|---|---|---|---|
| Age | 49.08 ± 10.31 years | 49.0 ± 11.73 years | 0.971 |
| Race | White (n = 26) | White (n = 61) | 0.205 |
|  | Asian (n = 3) | Asian (n = 7) |  |
|  | Black (n = 1) | Black (n = 8) |  |
|  | Unknown (n = 6) | Unknown (n = 5) |  |
| Lesion Type | Multiple masses (n = 16) | Multiple masses (n = 49) | 0.409 |
|  | Multiple NME (n = 2) | Multiple NME (n = 3) |  |
|  | Single mass (n = 16) | Single mass (n = 27) |  |
|  | Single NME (n = 2) | Single NME (n = 2) |  |
| Receptor Status | HR + / HER2 + (n = 8) | HR + / HER2 + (n = 12) | <0.001 |
|  | HR + / HER2 - (n = 8) | HR + / HER2 - (n = 43) |  |
|  | HR—/ HER2 - (n = 12) | HR—/ HER2 - (n = 24) |  |
|  | HR—/ HER2 + (n = 8) | HR—/ HER2 + (n = 2) |  |
| SBR Grade | III [High] (n = 22) | III [High] (n = 55) | 0.488 |
|  | II [Intermediate] (n = 13) | II [Intermediate] (n = 23) |  |
|  | I [Low] (n = 0) | I [Low] (n = 3) |  |
|  | Unknown (n = 0) | Unknown (n = 1) |  |
| Longest Diameter | 3.67 +/- 2.21 cm | 4.18 +/- 2.19 cm | 0.252 |

Distributions of patient characteristics, such as the distribution of lesion types and race, were compared using the χ2 test for homogeneity. Patient age and maximum tumor diameter distributions between classes were compared using a t-test. Investigation of the receptor status characteristics were done using 2 sample z-test of proportions. NME: non-mass enhancing; HR: hormone receptor; HER2: human epidermal growth factor receptor 2; pCR: complete pathological response; SBR: Scarff-Bloom-Richardson.

class having a greater proportion of patients that have HR+/HER2- status type (p<0.001) and a lesser proportion of patients with the HR-/HER2+ status type (p<0.001).

Fig 3A shows post-contrast DCE MRIs for a pCR patient and Fig 3B shows these MRIs for a non pCR patient at the pre-treatment, early treatment, and mid-treatment time points. The tumors were hyperintense relative to background tissue. Tumor of the non-pCR patient shrunk moderately whereas the tumor of the pCR patient shrunk markedly with time.

Fig 4A shows post-contrast DWI MRIs for a pCR patient and Fig 4B shows these MRIs for a non pCR patient. DWI signals of the tumors were hyperintense relative to background tissue. Tumor of the non-pCR patient shrunk moderately whereas the tumor of the pCR patient shrunk markedly with time.

Fig 5 describes the 10 most important texture features out of 372 features. All top 10 features were GLCM textural features derived from MRI data. In comparison, patient data scored far lower in terms of F-score. Patient age, race, and lesion type had F-scores of 1, 0, and 3 respectively. Receptor status, SBR grade, and longest diameter of the tumor had F-scores of 6, 3, and 3 respectively.

For pCR patients, regional tumor ADCs were $0.59\pm0.05$, $0.81\pm0.04$, and $1.11\pm0.06$ x$10^{-3}$ mm$^2$/s at pre-treatment, early treatment, and mid-treatment, respectively (Fig 6). For non-pCR patients, the corresponding tumor ADC values were $0.59\pm0.03$, $0.74\pm0.03$, and $0.91\pm0.05$ x$10^{-3}$ mm$^2$/s. ADC values for both PCR and non-PCR patients increased with treatment, with pCR patients showing a larger increase progressively.

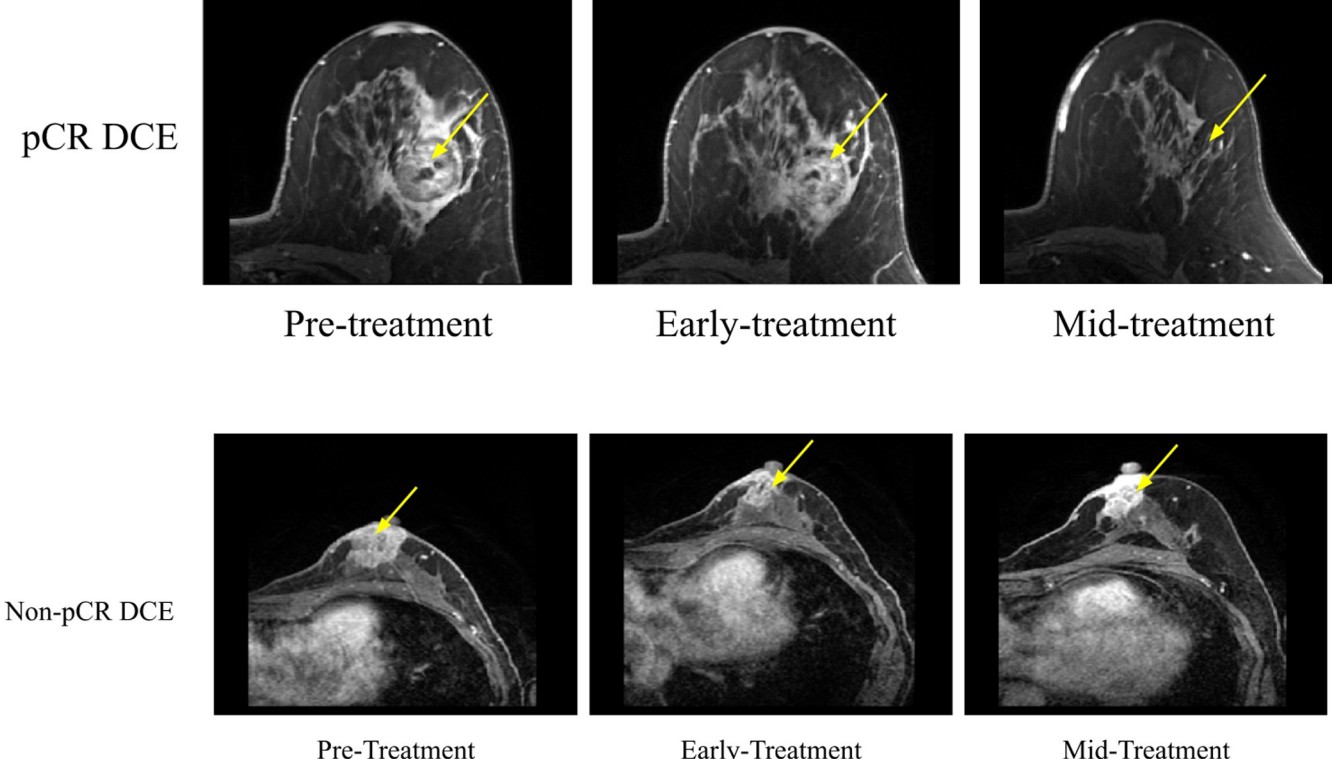

**Fig 3.** Post-contrast MRIs from DCE for (A) a pCR patient and (B) a non-PCR patient at the pre-treatment, early treatment, and mid-treatment time points. For (A), the 54 year old Caucasian patient had invasive ductal carcinoma with HR- and HER2-. For (B), the 51 year old Caucasian patient had invasive ductal carcinoma with HR+ and HER2-. The yellow arrow indicates the approximate location of the tumor. MRI images were taken approx. 2.5 minutes after contrast injection with a phase duration of 80sec < phase duration < 100sec. pCR: complete pathological response, DCE: dynamic contrast enhanced.

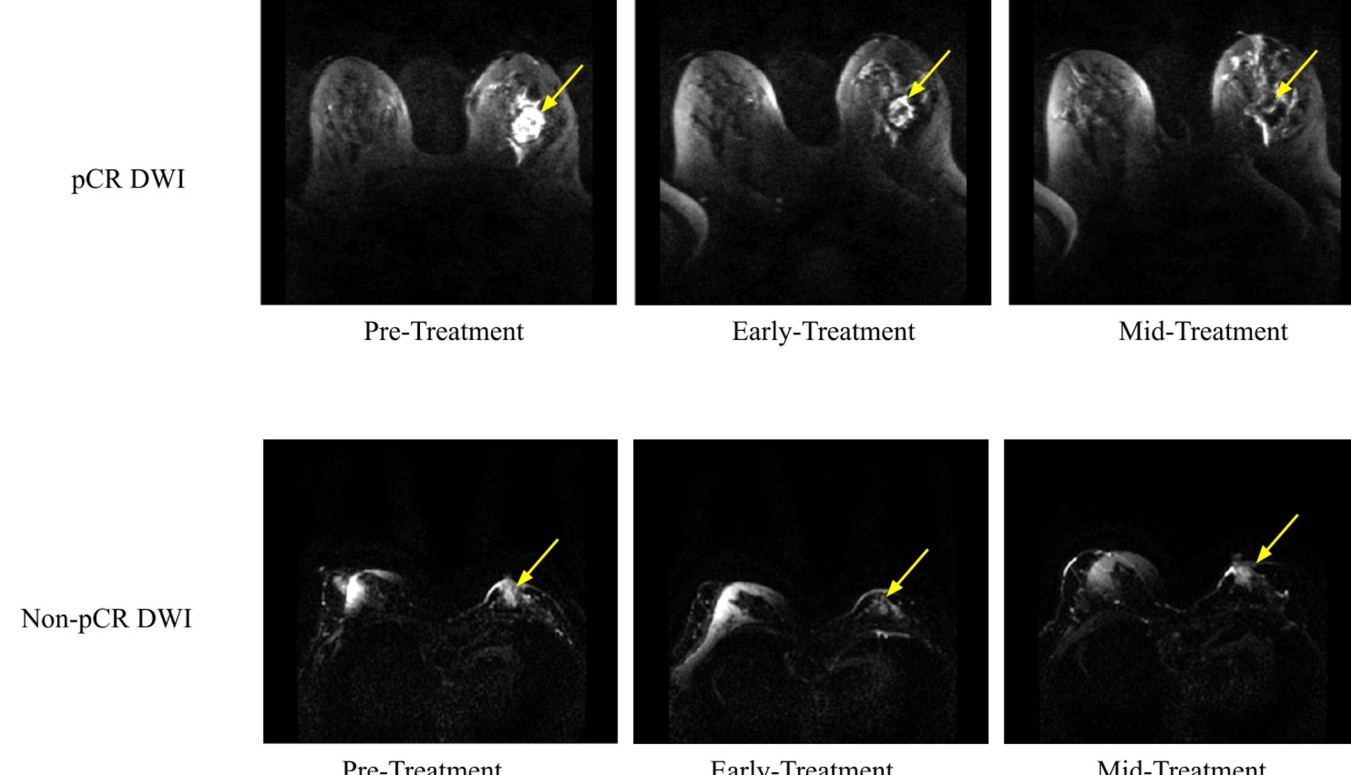

**Fig 4.** DWIs for (A) a pCR patient and (B) a non-PCR patient at the pre-treatment, early treatment, and mid-treatment time points. For (A), the 54 year old Caucasian patient had invasive ductal carcinoma with HR- and HER2-. For (B), the 51 year old Caucasian patient had invasive ductal carcinoma with HR+ and negative HER2-. The yellow arrow indicates the approximate location of the tumor. DWI acquisitions were done using a b-value of 800 s/mm$^2$. DWI: diffusion-weighted imaging).

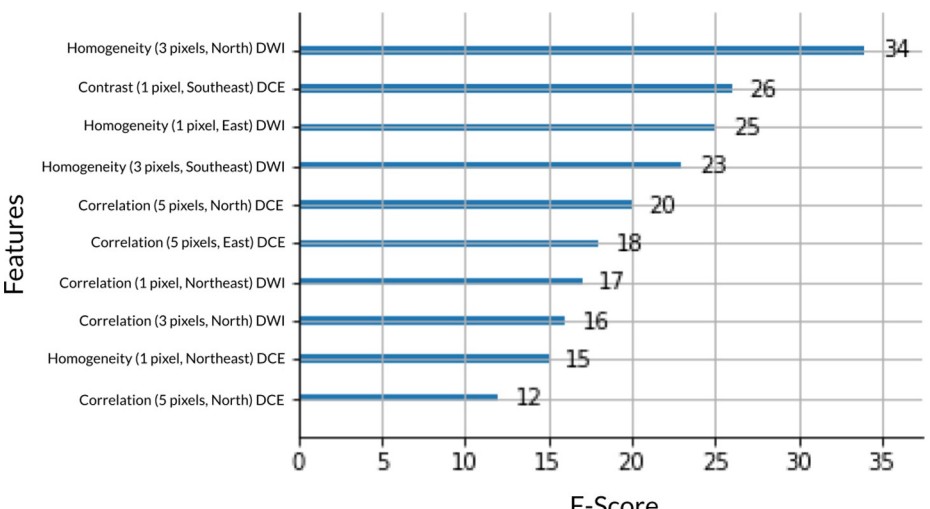

**Fig 5. Top 10 most important texture features out of 372 features by F-score.** GLCM features are shown with pixel distance value (1, 3 or 5 pixels), direction (North/South, East/West, North East/South West, or North West/South East), and MRI imaging type (DCE: dynamic contrast enhanced or DWI: diffusion-weighted imaging). GLCM: gray-level co-occurrence matrix.

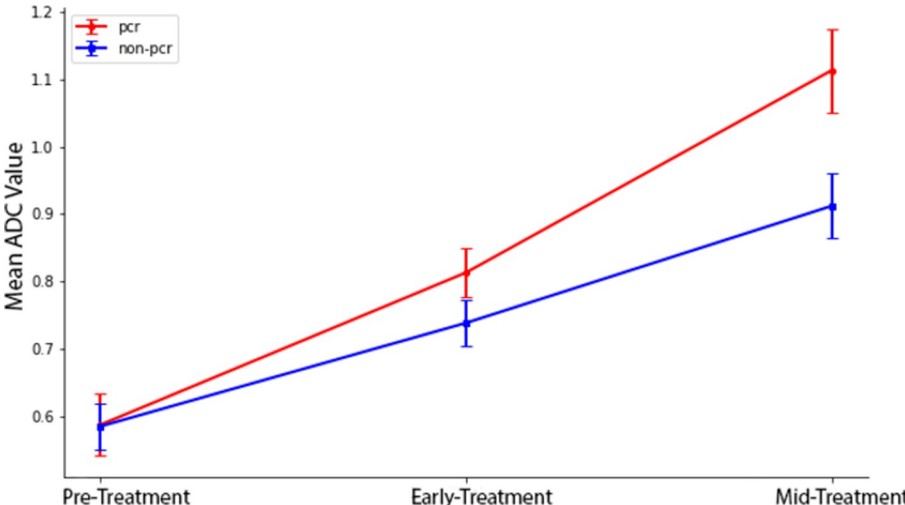

**Fig 6. Mean ADC values for all non-pCR and pCR patients at the pre-treatment, early treatment, and mid-treatment time points.** Error bars are standard deviations. pCR: complete pathological response.

## Prediction of pCR

Table 3 shows the AUC, precision, and recall performance of XGBoost for different data inputs after five-fold cross-validation. These AUC values were obtained using all treatment time points as input. With the mean tumor ADC, the AUC was 0.850 (95% CI: 0.764, 0.936; p<0.001). With texture features of DWI, the AUC was 0.871 (95% CI: 0.768, 0.974; p<0.001). With texture features of DCE images, the AUC was 0.903 (95% CI: 0.854, 0.952; p<0.001). Combining texture features of both DCE and DWI, the AUC was 0.916 (95% CI: 0.851, 0.981; p<0.001). All MRI data yielded an AUC of 0.933 (95% CI: 0.836, 1.03; p<0.001). By comparison, using non-imaging data yielded an AUC of 0.910 (95% CI: 0.848, 0.99; p<0.001). Prediction using all available imaging and non-imaging data yielded an AUC of 0.951 (95% CI: 0.909, 0.993; p<0.001).

We also evaluated the effects of using different combinations of treatment time points on prediction performance. For prediction models using all available MRI and non-MRI data, AUC using tp0, tp1, tp2, and tp0+tp1 were respectively, 0.918 (95% CI: 0.856, 0.98; p<0.001),

**Table 3. Model performances for models trained on ADC, DCE(GLCM), DWI(GLCM), their combinations, all non-imaging data, and the combination of all MRI and all non-MRI data with 5-fold cross-validation.**

| Features | AUC | Precision | Recall |
|---|---|---|---|
| ADC | 0.850 (0.764, 0.936; p<0.001) | 0.752 (0.666, 0.838; p<0.002) | 0.827 (0.753, 0.901; p<0.001)) |
| DWI GLCM | 0.871 (0.768, 0.974; p<0.001) | 0.779 (0.713, 0.845; p<0.001) | 0.926 (0.861, 0.991; p<0.001) |
| DCE GLCM | 0.903 (0.854, 0.952; p<0.001) | 0.856 (0.808, 0.904; p<0.001) | 0.939 (0.891, 0.987; p<0.001) |
| DCE+DWI GLCM | 0.916 (0.851, 0.981; p<0.001) | 0.779 (0.678, 0.880; p<0.003) | 0.915 (0.861, 0.969; p<0.001) |
| All MRI Data | 0.933 (0.836, 1.030; p<0.001) | 0.824 (0.780, 0.868; p<0.001) | 0.889 (0.848, 0.930; p<0.001) |
| All Non-MRI Data | 0.919 (0.848, 0.990; p<0.001) | 0.762 (0.665, 0.859; p<0.003) | 0.914 (0.888, 0.940; p<0.001) |
| All MRI + Non-MRI Data | 0.951 (0.909, 0.993; p<0.001) | 0.815 (0.690, 0.940; p<0.004) | 0.926 (0.844, 1.008; p<0.001) |

Values in parentheses are 95% confidence intervals. P-values are probability of AUC being 0.5. DWI: diffusion-weighted imaging; ADC: apparent diffusion coefficient; AUC: area under the curve; GLCM: gray-level co-occurrence matrix.

0.844 (95% CI: 0.705, 0.983; p = 0.004), 0.920 (95% CI: 0.858, 0.982; p<0.001), and 0.938 (95% CI: 0.89, 0.986; p<0.001). The AUC of using all time points were 0.951 (95% CI: 0.91, 0.992; p<0.001) (as shown in **Table 4**).

## Discussion

This study aimed to predict pCR non-invasively using XGBoost models with multiparametric MRI data at multiple treatment timepoints along with non-imaging data. This paper is the first to apply an extreme gradient boosting algorithm to the ISPY-2 data set to predict pathological complete response, and the second ever to use such an algorithm for predicting NAC response. This paper is also the first ever to use GLCM features to do so.

In the original I-SPY2 study, Partridge et al. [22] used logistic regression to evaluate if change in tumor ADC values (not texture analysis) is predictive of PCR in breast cancer patients. They found that changes in tumor ADC values was moderately predictive of pCR at mid-treatment (AUC = 0.60; 95% CI: 0.52, 0.68). In a test subset, a model combining tumor subtype and mid-treatment changes in ADC significantly improved predictive performance (AUC = 0.72; 95% CI: 0.61, 0.83).

There have only been a few studies that have applied deep learning on MR images themselves as inputs to predict pCR. Braman et al. [7] applied deep learning to predict NAC in HER2 patients from pre-treatment 2D DCE MRI in a retrospective study. They explored both pre-contrast and late post-contrast phases of DCE MRI and found an AUC of 0.74 (± 0.03). Qu et al. [24] built a CNN model using pre and post-NAC. Tumor regions were manually segmented by two expert radiologists on enhanced T1-weighted images. They found an AUC of 0.553 (95% CI: 0.416, 0.683) for pre-NAC data.

There have also only been a few studies that have applied non-deep learning on MR images themselves as inputs to predict pCR. Suo et al. [17] evaluated mono-exponential (ADC), bi-exponential and stretched-exponential from diffusion MRI data to predict pCR. They also included tumor size and relative enhancement ratio from DCE MRI data at 3 time points: before treatment, at mid-treatment, and after treatment with NAC. They found that flow-insensitive ADC change at mid-treatment was the most predictive feature, with an AUC of 0.831 (95% CI: 0.747, 0.915; $P < 0.001$). Combining this with receptor statuses, the AUC increased to 0.905 (95% CI: 0.843, 0.966; $P < 0.001$). Bian et al. [25] analyzed radiomic signatures based on T2W imaging, diffusion-weighted imaging, dynamic contrast-enhanced imaging and their combination to predict pCR. Logistic regression was then used to assess the association between features and clinical risk factors. The combined radiomic signature and nomogram model achieved an AUC of 0.91 (95% CI: 0.86, 1.00). Chen et al. [26] similarly analyzed the ability of radiomic signatures extracted from MRI data to predict pCR. They achieved an AUC of 0.848 by combining DCE-MRI and ADC maps. Eun et al. [19] performed non-GLCM texture analysis of pre and mid-treatment T2-weighted MRI, DCE, DWI, and ADC

**Table 4. Model performances for models trained on all features restricted to tp0, tp1, tp2, tp0 + tp1 and all timepoints using 5-fold cross-validation.**

| Timepoints | AUC | Precision | Recall |
|---|---|---|---|
| Tp0 | 0.918 (0.856, 0.980; p<0.001) | 0.778 (0.716, 0.840; p<0.001) | 0.913 (0.885, 0.941; p<0.001) |
| Tp1 | 0.844 (0.705, 0.983; p = 0.004) | 0.793 (0.653, 0.933; p = 0.007) | 0.841 (0.709, 0.973; p = 0.003) |
| Tp2 | 0.920 (0.858, 0.982; p<0.001) | 0.832 (0.77, 0.894; p<0.001) | 0.951 (0.929, 0.973; p<0.001) |
| Tp0 + Tp1 | 0.938 (0.89, 0.986; p<0.001) | 0.805 (0.757, 0.853; p<0.001) | 0.963 (0.918, 1.008; p<0.001) |
| Tp0 + Tp1 + Tp2 | 0.951 (0.910, 0.992; p<0.001) | 0.815 (0.690, 0.940; p = 0.004) | 0.926 (0.844, 1.008; p<0.001) |

Values in parentheses are 95% confidence intervals. P-values are probability of AUC being 0.5. AUC: area under the curve; Tp: time point.

mapping. The random forest classifier to predict PCR showed the highest diagnostic performance with mid-treatment DCE MRI (AUC = 0.82; 95% CI: 0.74, 0.88). Huang et al. [27] used a multilayer perceptron trained on radiomics features generated from MRI images (ADC maps, DCE, and fat-suppressed T2-weighted imaging), and reported an AUC of 0.900 AUC (95% CI: 0.849, 0.935) on the validation dataset. Tahmassebi et al. [20] was the only other paper to employ XGBoost to predict pCR. Using quantitative pharmacokinetic DCE features and ADC values in different classifier algorithms (support vector machine, linear discriminant analysis, logistic regression, random forests, stochastic gradient descent, decision tree, adaptive boosting and extreme gradient boosting), they achieved a final AUC of 0.86 with an extreme gradient boosting.

## Limitations

This study was performed on a relatively small multi-center dataset. As more data releases, our findings need to be replicated on a larger dataset to improve the generalizability of the model as well as account for the racial bias towards white women in this dataset. We have only evaluated XGBoost, other machine learning methods such as LightGBM should also be explored. Finally, we have only extracted GLCM features from MRI imaging data. Using additional features such as radiomic textural features may improve the model and give it more perspectives on the images. We employed images as provided and performed only visual image quality checks. We did not perform distortion correction, eddy current correction, check for multisite scanner consistency, among others, as raw data were not available. It has been shown that accurate estimates and reproducibility of diffusion indices in clinical studies require careful data quality assurance [28]. Moreover, large multicenter studies have shown a non-negligible variability in quantitative diffusion indices measured using different scanner systems [29, 30]. Therefore, MRI scanner system should be adequately characterized in diffusion-MRI of the breast [31]. Deep learning methods could also be applied to predict PCR [24, 32–37] (see review [38]) but were not analyzed.

## Conclusions

XGBoost models using multiparametric MRI data along with demographic and molecular subtype data, accurately predict pCR to NAC. With further development and testing on larger multi-institutional sample sizes, this approach has the potential to non-invasively identify patients who are likely to respond to neoadjuvant chemotherapy at diagnosis or early treatment. This approach may prove useful for treatment planning, treatment execution, and mid-treatment adjustment to achieve better outcomes.

## Author Contributions

**Conceptualization:** Aaquib Syed, Takouhie Maldjian, Tim Q. Duong.

**Data curation:** Aaquib Syed.

**Formal analysis:** Aaquib Syed, Tim Q. Duong.

**Funding acquisition:** Aaquib Syed.

**Investigation:** Aaquib Syed, Thomas Ren, Tim Q. Duong.

**Methodology:** Aaquib Syed, Takouhie Maldjian, Tim Q. Duong.

**Project administration:** Tim Q. Duong.

**Resources:** Tim Q. Duong.

**Software:** Aaquib Syed, Thomas Ren.

**Supervision:** Aaquib Syed, Richard Adam, Thomas Ren, Jinyu Lu, Takouhie Maldjian, Tim Q. Duong.

**Validation:** Aaquib Syed, Thomas Ren, Takouhie Maldjian, Tim Q. Duong.

**Visualization:** Aaquib Syed.

**Writing – original draft:** Aaquib Syed, Richard Adam, Jinyu Lu, Takouhie Maldjian, Tim Q. Duong.

**Writing – review & editing:** Aaquib Syed, Richard Adam, Jinyu Lu, Takouhie Maldjian, Tim Q. Duong.

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
