## [Decision Letter · Decision Letter 0]

25 Jul 2022

PONE-D-22-10717Machine learning with textural analysis of longitudinal multiparametric MRI and molecular subtypes accurately predicts Pathologic Complete Response in patients with invasive breast cancerPLOS ONE

Dear Dr. Duong,

Thank you for submitting your manuscript to PLOS ONE. After careful consideration, we feel that it has merit but does not fully meet PLOS ONE’s publication criteria as it currently stands. Therefore, we invite you to submit a revised version of the manuscript that addresses the points raised during the review process.

ACADEMIC EDITOR: This is a potentially interesting paper. However, the Authors should adequately address all the critical comments raised by the Reviewers. Moreover, they should discuss in greater detail the clinical impact of their findings. 

We look forward to receiving your revised manuscript.

Kind regards,

Marco Giannelli

Academic Editor

PLOS ONE

Journal Requirements:

Reviewers' comments:

Reviewer's Responses to Questions

**Comments to the Author**

1. Is the manuscript technically sound, and do the data support the conclusions?

Reviewer #1: Partly

Reviewer #2: Yes

2. Has the statistical analysis been performed appropriately and rigorously? 

Reviewer #1: I Don't Know

Reviewer #2: Yes

3. Have the authors made all data underlying the findings in their manuscript fully available?

Reviewer #1: Yes

Reviewer #2: Yes

4. Is the manuscript presented in an intelligible fashion and written in standard English?

Reviewer #1: Yes

Reviewer #2: Yes

5. Review Comments to the Author

Reviewer #1: 1) Given that DTI is a truly quantitative technique, a quality control of scanner performances in general and of correct application of diffusion weighting gradients in particular is needed, in order to guaranty accurate estimates of diffusion indices and reproducible results in clinical studies (Jones, Top Magn Reson Imaging 2010, 21: 87-99). Moreover, this is of paramount importance when pooling data from different scanners. Indeed, previous large multicenter studies have shown a non-negligible variability in quantitative diffusion indices measured using different scanner systems (Fedeli et al, Phys Med 2018, 55: 135-141; Fedeli et al, Phys Med 2021, 85: 98-106). Therefore, MR scanner system should be adequately characterized in diffusion-MRI of the breast (Giannelli et al, PLoS One 2014, 9: e86280). The Authors should discuss in detail these aspects.

2) The Authors have focused on a single feature class, i.e., GLCM. The Authors should explain this choice and why they have not considered additional feature classes.

3) When applying machine learning techniques on multicenter data, it would be appropriate using methods of data harmonization (Johnson et al, Biostatistics 2007, 8: 118-127). The Authors should hence discuss this limitation of their study.

4) Given the sensitivity of radiomic features to image resampling and discretization, the Authors should report the resampling voxel size and bin width/number of bins used for the preprocessing of radiomic data, discussing whether the used values are optimal.

Reviewer #2: In this retrospective paper, the authors were able to predict pCR after neoadjuvant chemotherapy using XGBoost with MRI and non-imaging data at multiple treatment timepoints.

1. In Abstract Purpose and throughout the paper, I suggest pathological complete response (pCR) and extreme gradient boosting (XGBoost).

2. In Abstract Results, 1st sentence, was 0.903 an AUC value?

3. In Introduction, 1st paragraph, last sentence, more details are needed.

4. In Introduction, 2nd paragraph, 1st sentence, a should be added before chemotherapy.

5. In Introduction, 3rd paragraph, 2nd sentence, see reviews can be cut and and should be added before DCE dynamics.

6. In Methodology, Data Sources, 1st paragraph, penultimate sentence, I suggest logistic regression.

7. In Methodology, Data Sources, 2nd paragraph, 1st sentence, and throughout the paper, I suggest paclitaxel and anthracycline.

8. In Methodology, Ground Truth, 1st sentence, FTV is not clear.

9. In Methodology, Statistical Analysis, 3rd paragraph, line 2, Recall should be recall. In line 4, the material in parentheses, starting with model, can be deleted.

10. In Results, 1st paragraph, 2nd sentence, the should be added before mean.

11. In Discussion, 4th paragraph, line 8, derived from should be cut.

12. In Conclusion, 1st sentence, subtypes should be changed to subtype. In the 2nd sentence, a should be added after predict.

13. The references should be written according to the journal's style.

14. In the figures, I suggest one patient per figure, and the pulse sequences should be described in the legends

6. PLOS authors have the option to publish the peer review history of their article (what does this mean?). If published, this will include your full peer review and any attached files.

Reviewer #1: No

Reviewer #2: **Yes: **Gary J. Whitman MD

---

## [Author Response · Author response to Decision Letter 0]

7 Aug 2022

All responses are included in the "Response to Reviews" file. Below is a copy of the document:

PONE-D-22-10717

Dear Editors and Reviewers, 

We thank you for your thoughtful reviews of our manuscript. The point by point response and revised manuscript are attached. Thank you. 

Tim Duong on behalf of coauthors. 

Machine learning with textural analysis of longitudinal multiparametric MRI and molecular subtypes accurately predicts Pathologic Complete Response in patients with invasive breast cancer

PLOS ONE

ACADEMIC EDITOR:

This is a potentially interesting paper. However, the Authors should adequately address all the critical comments raised by the Reviewers. Moreover, they should discuss in greater detail the clinical impact of their findings. 

- Thank you. The followings have been added: 

- With further development and testing on larger multi-institutional sample sizes, this approach has the potential to non-invasively identify patients who are likely to respond to neoadjuvant chemotherapy at diagnosis or early treatment. This approach may prove useful for treatment planning, treatment execution, and mid-treatment adjustment to achieve better outcomes.

Reviewers' comments:

Reviewer's Responses to Questions 

Comments to the Author

1. Is the manuscript technically sound, and do the data support the conclusions?

Reviewer #1: Partly

Reviewer #2: Yes

2. Has the statistical analysis been performed appropriately and rigorously? 

Reviewer #1: I Don't Know

- We expanded the section on Methods to better describe the statistical tests used. We also include statistical tests used in the results and legends where appropriate or unclear. 

Reviewer #2: Yes

3. Have the authors made all data underlying the findings in their manuscript fully available?

Reviewer #1: Yes

Reviewer #2: Yes

4. Is the manuscript presented in an intelligible fashion and written in standard English?

Reviewer #1: Yes

Reviewer #2: Yes

5. Review Comments to the Author

Reviewer #1: 1) Given that DTI is a truly quantitative technique, a quality control of scanner performances in general and of correct application of diffusion weighting gradients in particular is needed, in order to guaranty accurate estimates of diffusion indices and reproducible results in clinical studies (Jones, Top Magn Reson Imaging 2010, 21: 87-99). Moreover, this is of paramount importance when pooling data from different scanners. Indeed, previous large multicenter studies have shown a non-negligible variability in quantitative diffusion indices measured using different scanner systems (Fedeli et al, Phys Med 2018, 55: 135-141; Fedeli et al, Phys Med 2021, 85: 98-106). Therefore, MR scanner system should be adequately characterized in diffusion-MRI of the breast (Giannelli et al, PLoS One 2014, 9: e86280). The Authors should discuss in detail these aspects.

We completely agree with these comments. Raw images are not available to make corrections. Per open data sources, these images underwent reasonable and vigorous data quality check, , ranging from distortion correction, eddy current correction, quality assurance with phantom and human data, pooling data from multiple sites. 

- Thank you for the references. They are cited. The following is added: 

- We only performed visual check on images as provided but did not perform distortion correction, eddy current correction, check for multisite scanner consistency, among others, as raw data were not available. It has been shown that accurate estimates and reproducibility of diffusion indices in clinical studies require careful data quality assurance (Jones, Top Magn Reson Imaging 2010, 21: 87-99). Moreover, large multicenter studies have shown a non-negligible variability in quantitative diffusion indices measured using different scanner systems (Fedeli et al, Phys Med 2018, 55: 135-141; Fedeli et al, Phys Med 2021, 85: 98-106). Thus, MRI scanner system should be adequately characterized in diffusion-MRI of the breast (Giannelli et al, PLoS One 2014, 9: e86280). 

2) The Authors have focused on a single feature class, i.e., GLCM. The Authors should explain this choice and why they have not considered additional feature classes.

- Thank you for your comment. The following is added. 

- While there are many texture features have been explored, we only chose GLCM features because they haven't been widely used before compared to other radiomic features. Future studies should compare results from different texture features. 

3) When applying machine learning techniques on multicenter data, it would be appropriate using methods of data harmonization (Johnson et al, Biostatistics 2007, 8: 118-127). The Authors should hence discuss this limitation of their study.

- Thank you for the references. They are cited. The following is added:

- When applying machine learning techniques on multicenter data, data harmonization (Johnson et al, Biostatistics 2007, 8: 118-127) may be necessary. However, we did not perform additional data harmonization as all MRI acquisitions were taken with ISPY2 acquisition requirements as described in (20). Furthermore, we ensured that data from different field strengths and different molecular subtypes, etc., were not over-represented in either training and testing data sets by using 5-fold cross validation so that all data has the chance to be in both the test and training sets for a given fold.

4) Given the sensitivity of radiomic features to image resampling and discretization, the Authors should report the resampling voxel size and bin width/number of bins used for the preprocessing of radiomic data, discussing whether the used values are optimal.

- Thank you for the comment. The following is added:

- All images were interpolated to 0.7825 mm x 0.7825 mm voxel spacing with a 2.5 mm slice thickness, which was the median spacing and thickness for all patients. Images were cast to 8-bits, and the number of bins was set to the maximum value of 256 to maximize the number of grey levels counted. 

Reviewer #2: In this retrospective paper, the authors were able to predict pCR after neoadjuvant chemotherapy using XGBoost with MRI and non-imaging data at multiple treatment timepoints.

- For all comments below, all were revised as suggested (but not marked) except where details are noted: 

1. In Abstract Purpose and throughout the paper, I suggest pathological complete response (pCR) and extreme gradient boosting (XGBoost).

2. In Abstract Results, 1st sentence, was 0.903 an AUC value?

3. In Introduction, 1st paragraph, last sentence, more details are needed.

- It is revised as:

- Thus, accurate prediction of pCR could help treatment planning, identifying patients who are likely to respond to specific NAC drugs , and alter treatments mid-course if needed, to maximize successful outcomes while minimize adverse effects of unnecessary chemotherapy.

4. In Introduction, 2nd paragraph, 1st sentence, a should be added before chemotherapy.

5. In Introduction, 3rd paragraph, 2nd sentence, see reviews can be cut and and should be added before DCE dynamics.

6. In Methodology, Data Sources, 1st paragraph, penultimate sentence, I suggest logistic regression.

7. In Methodology, Data Sources, 2nd paragraph, 1st sentence, and throughout the paper, I suggest paclitaxel and anthracycline.

8. In Methodology, Ground Truth, 1st sentence, FTV is not clear.

9. In Methodology, Statistical Analysis, 3rd paragraph, line 2, Recall should be recall. In line 4, the material in parentheses, starting with model, can be deleted.

10. In Results, 1st paragraph, 2nd sentence, the should be added before mean.

11. In Discussion, 4th paragraph, line 8, derived from should be cut.

12. In Conclusion, 1st sentence, subtypes should be changed to subtype. In the 2nd sentence, a should be added after predict.

13. The references should be written according to the journal's style.

14. In the figures, I suggest one patient per figure, and the pulse sequences should be described in the legends

- Thank you. all are revised as suggested.

---

## [Decision Letter · Decision Letter 1]

3 Oct 2022

PONE-D-22-10717R1Machine learning with textural analysis of longitudinal multiparametric MRI and molecular subtypes accurately predicts Pathologic Complete Response in patients with invasive breast cancerPLOS ONE

Dear Dr. Duong,

Thank you for submitting your manuscript to PLOS ONE. After careful consideration, we feel that it has merit but does not fully meet PLOS ONE’s publication criteria as it currently stands. Therefore, we invite you to submit a revised version of the manuscript that addresses the points raised during the review process.

We look forward to receiving your revised manuscript.

Kind regards,

Marco Giannelli

Academic Editor

PLOS ONE

Journal Requirements:

Additional Editor Comments:

The Authors have improved the manuscript, which however present some remaining minor concerns to be addressed.

Reviewers' comments:

Reviewer's Responses to Questions

**Comments to the Author**

1. If the authors have adequately addressed your comments raised in a previous round of review and you feel that this manuscript is now acceptable for publication, you may indicate that here to bypass the “Comments to the Author” section, enter your conflict of interest statement in the “Confidential to Editor” section, and submit your "Accept" recommendation.

Reviewer #2: (No Response)

2. Is the manuscript technically sound, and do the data support the conclusions?

Reviewer #2: Yes

3. Has the statistical analysis been performed appropriately and rigorously? 

Reviewer #2: Yes

4. Have the authors made all data underlying the findings in their manuscript fully available?

Reviewer #2: Yes

5. Is the manuscript presented in an intelligible fashion and written in standard English?

Reviewer #2: Yes

6. Review Comments to the Author

Reviewer #2: This paper on machine learning in predicting pathologic complete response in patients with invasive breast cancer has been improved since an earlier submission.

1. In the Introduction, 1st paragraph, line 8, regimen should be changed to regimens.

2. In the Introduction, 2nd paragraph, the 2nd sentence should start with Clinical biomarkers.

3. In the Introduction, 3rd paragraph, last sentence, the should be added before results.

4. In Materials and Methods, Data Sources, 4th paragraph, 1st sentence, I suggest: MRI studies, including DWI and DCE MRI, were. In the last sentence of that paragraph, for a given fold is not clear.

5. Regarding the 1st table in the Results section, the following abbreviations should be defined in a key: NME, HR, HER2, pCR, and SBR.

6. In Table 3, the following abbreviations should be defined in a key: DWI, ADC, AUC, and GLCM.

7. In Table 4, the following abbreviations should be defined in a key: AUC and Tp.

8. In Limitations, line 7, check should be checks.

9. The references should be written according to the journal's style. In references 3, 6, 16, 18-20, 22, 26, 27, and 31, only the first word of the title of the article should be capitalized.

10. Figure legends should be added.

7. PLOS authors have the option to publish the peer review history of their article (what does this mean?). If published, this will include your full peer review and any attached files.

Reviewer #2: **Yes: **Gary J. Whitman

---

## [Author Response · Author response to Decision Letter 1]

5 Oct 2022

Reviewer #2: This paper on machine learning in predicting pathologic complete response in patients with invasive breast cancer has been improved since an earlier submission.

1. In the Introduction, 1st paragraph, line 8, regimen should be changed to regimens.

2. In the Introduction, 2nd paragraph, the 2nd sentence should start with Clinical biomarkers.

3. In the Introduction, 3rd paragraph, last sentence, the should be added before results.

4. In Materials and Methods, Data Sources, 4th paragraph, 1st sentence, I suggest: MRI studies, including DWI and DCE MRI, were. In the last sentence of that paragraph, for a given fold is not clear.

5. Regarding the 1st table in the Results section, the following abbreviations should be defined in a key: NME, HR, HER2, pCR, and SBR.

6. In Table 3, the following abbreviations should be defined in a key: DWI, ADC, AUC, and GLCM.

7. In Table 4, the following abbreviations should be defined in a key: AUC and Tp.

8. In Limitations, line 7, check should be checks.

9. The references should be written according to the journal's style. In references 3, 6, 16, 18-20, 22, 26, 27, and 31, only the first word of the title of the article should be capitalized.

10. Figure legends should be added.

ALL comments were incorporated exactly as suggested. Thank you.

---

## [Decision Letter · Decision Letter 2]

19 Dec 2022

PONE-D-22-10717R2Machine learning with textural analysis of longitudinal multiparametric MRI and molecular subtypes accurately predicts Pathologic Complete Response in patients with invasive breast cancerPLOS ONE

Dear Dr. Duong,

Thank you for submitting your manuscript to PLOS ONE. After careful consideration, we feel that it has merit but does not fully meet PLOS ONE’s publication criteria as it currently stands. Therefore, we invite you to submit a revised version of the manuscript that addresses the points raised during the review process.

We look forward to receiving your revised manuscript.

Kind regards,

Marco Giannelli

Academic Editor

PLOS ONE

Journal Requirements:

Additional Editor Comments:

The Reviewer has proposed some suggestions in order to definitively improve the manuscript. I hope the Authors are willing to accomodate these constructive suggestions.

Reviewers' comments:

Reviewer's Responses to Questions

**Comments to the Author**

1. If the authors have adequately addressed your comments raised in a previous round of review and you feel that this manuscript is now acceptable for publication, you may indicate that here to bypass the “Comments to the Author” section, enter your conflict of interest statement in the “Confidential to Editor” section, and submit your "Accept" recommendation.

Reviewer #2: (No Response)

2. Is the manuscript technically sound, and do the data support the conclusions?

Reviewer #2: Yes

3. Has the statistical analysis been performed appropriately and rigorously? 

Reviewer #2: Yes

4. Have the authors made all data underlying the findings in their manuscript fully available?

Reviewer #2: Yes

5. Is the manuscript presented in an intelligible fashion and written in standard English?

Reviewer #2: Yes

6. Review Comments to the Author

Reviewer #2: This paper on assessing response to neoadjuvant chemotherapy in breast cancer with MRI has been improved since an earlier submission.

1. In the 1st paragraph of the Introduction, 2nd sentence, I suggest: Pathological complete response...tumor present at surgery post NAC.

2. In the fourth paragraph of the Introduction, the 2nd sentence should start with Extreme gradient boosting.

3. In Materials and Methods, Data Sources, last sentence of the last paragraph, I suggest: data had the...test and the training sets.

4. In Materials and Methods, Ground Truth, first sentence, I suggest: functional tumor volume.

5. In Results, Prediction of pCR, last paragraph, 2nd sentence, model should be models.

6. In Discussion, 4th paragraph, 6th sentence, from should be deleted.

7. In Limitations, 4th sentence, image should be images.

8. The references should be written according to the journal's style. The journal names should be capitalized and abbreviations should be used. In reference 16, in the title of the chapter, rrimary should be primary, and the title of the book should start with Breast. In reference 19, Association should be association.

9. I suggest one patient per figure. In the figure legends, I suggest including the age of the patient and the breast tumor pathologic diagnosis.

7. PLOS authors have the option to publish the peer review history of their article (what does this mean?). If published, this will include your full peer review and any attached files.

Reviewer #2: **Yes: **Gary J. Whitman, MD

---

## [Author Response · Author response to Decision Letter 2]

22 Dec 2022

R3

Dear Dr. Giannelli and reviewers, 

Thank you for your careful reading of the manuscript. 

Thank you for the grammatic corrections.

Please find the point-by-point responses and the revised manuscript. 

I wish you and your family a happy holiday season. 

Sincerely, Tim Duong. 

Reviewer #2: This paper on assessing response to neoadjuvant chemotherapy in breast cancer with MRI has been improved since an earlier submission.

1. In the 1st paragraph of the Introduction, 2nd sentence, I suggest: Pathological complete response...tumor present at surgery post NAC.

2. In the fourth paragraph of the Introduction, the 2nd sentence should start with Extreme gradient boosting.

3. In Materials and Methods, Data Sources, last sentence of the last paragraph, I suggest: data had the...test and the training sets.

4. In Materials and Methods, Ground Truth, first sentence, I suggest: functional tumor volume.

5. In Results, Prediction of pCR, last paragraph, 2nd sentence, model should be models.

6. In Discussion, 4th paragraph, 6th sentence, from should be deleted.

7. In Limitations, 4th sentence, image should be images.

8. The references should be written according to the journal's style. The journal names should be capitalized and abbreviations should be used. In reference 16, in the title of the chapter, rrimary should be primary, and the title of the book should start with Breast. In reference 19, Association should be association.

Thank you for all the grammatic corrections above. Changes are not tracked. 

9. I suggest one patient per figure. In the figure legends, I suggest including the age of the patient and the breast tumor pathologic diagnosis.

This information is provided as suggested. 

Fig 3. Post-contrast MRIs from DCE for (A) a pCR patient and (B) a non-PCR patient at the pre-treatment, early treatment, and mid-treatment time points. For (A), the 54 year old Caucasian patient had invasive ductal carcinoma with HR- and HER2-. For (B), the 51 year old Caucasian patient had invasive ductal carcinoma with HR+ and HER2-. The yellow arrow indicates the approximate location of the tumor. MRI images were taken approx. 2.5 minutes after contrast injection with a phase duration of 80sec < phase duration < 100sec. pCR: complete pathological response, DCE: dynamic contrast enhanced.

Fig 4. DWIs for (A) a pCR patient and (B) a non-PCR patient at the pre-treatment, early treatment, and mid-treatment time points. For (A), the 54 year old Caucasian patient had invasive ductal carcinoma with HR- and HER2-. For (B), the 51 year old Caucasian patient had invasive ductal carcinoma with HR+ and negative HER2-. The yellow arrow indicates the approximate location of the tumor. DWI acquisitions were done using a b-value of 800 s/mm2. DWI: diffusion-weighted imaging).

---

## [Editor Report · Decision Letter 3]

27 Dec 2022

Machine learning with textural analysis of longitudinal multiparametric MRI and molecular subtypes accurately predicts Pathologic Complete Response in patients with invasive breast cancer

PONE-D-22-10717R3

Dear Dr. Duong,

We’re pleased to inform you that your manuscript has been judged scientifically suitable for publication and will be formally accepted for publication once it meets all outstanding technical requirements.

Kind regards,

Marco Giannelli

Academic Editor

PLOS ONE

Additional Editor Comments (optional):

The Authors have adequately addressed all the concerns raised by the Reviewers.
---

## [Editor Report · Acceptance letter]

3 Jan 2023

PONE-D-22-10717R3 

Machine learning with textural analysis of longitudinal multiparametric MRI and molecular subtypes accurately predicts Pathologic Complete Response in patients with invasive breast cancer 

Dear Dr. Duong:

I'm pleased to inform you that your manuscript has been deemed suitable for publication in PLOS ONE. Congratulations! Your manuscript is now with our production department. 

Kind regards, 

on behalf of

Dr. Marco Giannelli 

Academic Editor

PLOS ONE